# Correlations between Peak Nasal Inspiratory Flow, Acoustic Rhinometry, 4-Phase Rhinomanometry and Reported Nasal Symptoms

**DOI:** 10.3390/jpm12091513

**Published:** 2022-09-15

**Authors:** Giancarlo Ottaviano, Alfonso Luca Pendolino, Bruno Scarpa, Miriam Torsello, Daniele Sartori, Enrico Savietto, Elena Cantone, Piero Nicolai

**Affiliations:** 1Department of Neurosciences, Otolaryngology Section, University of Padova, 35121 Padova, Italy; 2Department of ENT, Royal National ENT & Eastman Dental Hospitals, London WC1E 6DG, UK; 3Ear Institute, University College London, London WC1X 8EE, UK; 4Department of Statistical Sciences and Department of Mathematics Tullio Levi-Civita, University of Padova, 35128 Padova, Italy; 5Department of Neurosciences, Reproductive and Odontostomatologic Sciences, Unit of Ear, Nose and Throat, Federico II University, 80131 Naples, Italy

**Keywords:** PNIF, 4-phase rhinomanometry, acoustic rhinometry, VAS, SNOT-22, nasal obstruction, nasal symptoms, quality of life, questionnaire, objective measurements

## Abstract

Background: Rhinomanometry, acoustic rhinometry (AR) and peak nasal inspiratory flow (PNIF) are popular methods for nasal patency evaluation. The aim of the present study was to compare these three methods with the reported nasal symptoms to determine the best diagnostic tool to assess nasal obstruction. Methods: 101 subjects were evaluated using PNIF, 4-phase rhinomanometry (4PR), AR, Visual Analogue Scale for nasal obstruction (VAS-NO) and Sino-Nasal Outcome Test (SNOT-22). Correlations among PNIF, 4PR, AR, VAS-NO and SNOT-22 were obtained. Results: VAS-NO and SNOT-22 were moderately correlated with each other (r = 0.54, *p* < 0.001). 4PR was moderately correlated with PNIF (r = –0.31, *p* = 0.0016) and AR (r = –0.5, *p* < 0.001). VAS-NO was mildly correlated with PNIF (r = –0.29, *p* = 0.0034). SNOT-22 was moderately correlated with PNIF (r = –0.31, *p* = 0.0017). After dividing the population into symptomatic and asymptomatic subjects, based on their VAS-NO score, the former showed significantly lower PNIF values (*p* = 0.009) and higher 4PR values (*p* = 0.013) compared to the latter ones. Conclusion: PNIF and 4PR showed a significant moderate correlation with each other, but PNIF showed a significant correlation (weak-moderate) with the reported nasal symptom scores.

## 1. Introduction

Nasal obstruction is one of the most common rhinologic symptoms, being reported by up to 80% of ENT patients. It can be bilateral or unilateral, constant or alternating. While a constant, unilateral nasal blockage is usually related to a septal deviation or a sinonasal mass, alternating/fluctuating nasal obstruction is more often associated with rhinitis [1]. 

Rhinomanometry (RM), acoustic rhinometry (AR) and peak nasal inspiratory flow (PNIF) are among the most popular tools that are used to objectively assess nasal obstruction. Whereas RM and PNIF measure nasal airflow, respectively with or without simultaneous intranasal pressure recording, AR assesses the nasal volume and the cross-sectional area. Several studies have compared these three methods in the evaluation of nasal obstruction, but the results are conflicting. Some authors have reported a moderate-to-strong correlation between PNIF and both AR and RM [2,3], while others did not find any association between PNIF and RM [4]. Current guidelines recommend the use of four-phase rhinomanometry (4PR) for the assessment of nasal obstruction. In this method, nasal airway resistance is calculated using hundreds of resistances which are continuously recorded during the whole breathing cycle [5]. 

On the other hand, subjective nasal obstruction can be evaluated using patient reported outcome measures (PROMs) such as the Visual Analogue Scale for nasal obstruction (VAS-NO). However, the correlation between subjective sensation of nasal obstruction and its objective measurement is not clear [6]. While some authors found no correlation between VAS-NO and RM [7] or AR [8,9], others have showed a good correlation between VAS-NO and RM [10] in patients with either allergic or non-allergic rhinitis [11]. Recently, a correlation between VAS-NO and PNIF has also been demonstrated [12]. 

In the last decade, precision medicine has revolutionized the clinical practice [13]. When this is applied to the assessment of the blocked nose, a comprehensive evaluation of nasal patency remains extremely important [3]. However, this has always been a challenge given the limited diffusion of nasal obstruction measurements [14] and their poor correlation with subjective nasal obstruction [6]. 

The aim of the present cross-sectional study was to compare PNIF, AR and 4PR in a group of adult subjects and to evaluate their correlation with reported nasal symptoms measured by means of VAS-NO and SNOT-22 in order to advise on the best diagnostic method to use in the clinical practice for nasal obstruction assessment.

## 2. Materials and Methods

One hundred and one consecutive subjects (58 males and 43 females) were enrolled between March and June 2019 at our ENT Clinic, Department of Neurosciences, University Hospital of Padova. Subjects that were under 18 years, unable to give informed consent, unwilling to participate or lacking fluency in the Italian language were excluded. The present investigation was conducted in accordance with the principles of 1996 Helsinki Declaration. All patients signed a written permission form for the clinical publication of the data. Data were examined in agreement with the Italian privacy and sensible data laws (D.Lgs 196/03). The study was approved by Padua Otolaryngology Section’s ethical committee (Prot. n. 3947/16). 

After collecting the subjects’ age, BMI, allergy status, asthma and smoking status at the time of enrollment, all subjects completed a VAS-NO (where 0 corresponded to not affected and 10 corresponded to the worst thinkable situation) and a SNOT-22 questionnaire [15]. PNIF, AR and 4PR were performed on the same occasion to objectively assess their nasal obstruction. PNIF was measured using a portable Youlten peak flow meter (Clement Clark International). Three maximal inspirations were obtained and the highest of the three measurements was considered [16]. Unilateral PNIF values were also studied by sealing off one nostril at a time with adhesive tape (Microfoam™, 3M™), and the highest values were taken as left PNIF (lPNIF) and right PNIF (rPNIF) [17]. AR was performed using the A1 acoustic rhinometer (GM Instruments Ltd., Kilwinning, UK) and conducted during breath holding. The minimal cross-sectional area (MCA) was evaluated [18]. Nasal airway resistances were measured with 4PR (4RHINO, Rhinolab, Freiburg, Germany) and the logarithmic effective resistances (logReff) during inspiration were considered, as reported in the literature [5]. 

All the measurements were performed on the patients when they were in a sitting position [19] by the same operator (DS), after at least 10 min of acclimatization in a room with constant temperature (between 19 and 22 °C) and a relative humidity of 25–35%. The subjects were instructed to avoid excessive smoking and caffeine intake during the day before the evaluation, but none were requested to modify their usual medication intake, in order to reflect the most common state that was experienced by the patients in their daily life [20].

### Statistical Analysis

The Bravais-Pearson correlation coefficient was used to measure the relationship between the different indicators. In order to adjust for sex, age, BMI, allergy, asthma and smoking, different linear models were fitted using PNIF, AR, 4PR, VAS-NO, SNOT-22 as response variable; correlations among residuals were used as measures of the adjusted correlation.

*p*-values were calculated for all tests, and 5% was considered as the critical level of significance. Assuming that data are normally distributed, with a correlation r = 0.3, a power of 0.8 was obtained for 84 subjects, with a 5% level of significance (alpha = 0.05). 

The R: a language and environment for statistical computing (R Foundation for Statistical Computing, Vienna, Austria) was used for all analyses including the power calculation. The STROBE guidelines [21] for cross-sectional studies were followed in the reporting of this study.

## 3. Results

Demographic and clinical data are summarized in Table 1. 

Among the 101 subjects that were enrolled, 80 were referred for sinonasal problems, while the remaining had no sinonasal concerns and were referred for other clinical reasons. Of the 80 patients with a nasal pathology, two did not fill the VAS-NO and SNOT-22 questionnaires but were still included in the statistical analysis to study the correlations between PNIF, AR and 4PR.

VAS-NO and SNOT-22 showed a positive correlation (r = 0.54, *p* < 0.001). PNIF was negatively correlated with 4PR (r = −0.31, *p* = 0.0016), SNOT-22 (r = −0.31, *p* = 0.0017) and VAS-NO (r = −0.29, *p* = 0.0034). 4PR was negatively correlated with AR (r = −0.5, *p* < 0.0001). lPNIF was positively correlated with lAR (r = 0.26, *p* = 0.0087) and negatively with l4PR (r = −0.3 *p* = 0.0025) and VAS-NO (r = −0.26, *p* = 0.0097). lAR was negatively correlated with l4PR (r = −0.51, *p* < 0.001). rPNIF was positively correlated with rAR (r = 0.32, *p* = 0.0011) and negatively with r4PR (r = −0.35, *p* < 0.001) and SNOT-22 (r = −0.22, *p* = 0.028). rAR was negatively correlated with r4PR (r = −0.46 *p* < 0.0001). No other correlations among the three different methods and between these and PROMs were observed. (Figure 1, Figure 2 and Figure 3).

Once we considered the effects of sex, age, BMI, allergy and smoking status, no relevant differences were observed in the results.

After grouping the subjects into those that were complaining of a moderate-to-severe nasal obstruction (VAS-NO ≥ 5) and those reporting an absent or a mild nasal obstruction (VAS-NO < 5) [22], we found that PNIF was significantly lower (*p* = 0.009) and 4PR significantly higher (*p* = 0.013) in the former group. AR did not show any significant difference between the two groups (Figure 4a). When we divided the subjects into those with sinonasal symptoms (SNOT-22 ≥ 22) and those without (SNOT-22 < 22) [23,24], neither PNIF nor 4PR or AR showed any significant difference (Figure 4b).

## 4. Discussion

Nasal obstruction is a multifactorial symptom as it could be a consequence of the intranasal anatomical status, but it is also influenced by the autonomic nervous system and other different physiological and pathological factors [25]. Additionally, it has been shown that its estimation can be influenced by patients’ psychological status or expectations; thus, it can be reported even in the absence of a genuine objective nasal obstruction [26].

There is still an open debate on the best method to use to assess nasal patency. RM has been considered for many years the gold standard for the measurement of nasal obstruction, with anterior active rhinomanometry being widely used as the preferred method [1,5]. More recently, 4PR has been introduced as a new method that is able to measure nasal resistance in the four different phases of the breathing cycle (ascending and descending phases both in inspiration and expiration) [1]. PNIF and AR are well-known and reliable methods to assess nasal airflow and patency in both healthy and obstructed noses [1]. However, it has been suggested that AR, by measuring different nasal parameters (nasal volume and MCA), should be considered as a complementary technique to PNIF and/or RM [27]. Recently, van Egmond and coworkers [28], comparing 4PR and PNIF with the subjective sensation of nasal obstruction in 111 patients, found both instruments to be comparable. The authors concluded that PNIF could be a better choice for its user-friendliness and practical advantages over 4PR [28]. Some years ago, Numminen and coworkers [29] compared RM, AR, PNIF with VAS and observed that these methods were correlate with each other. The authors concluded that these four methods support each other well in pathological noses and allow for the identification of sensitive intranasal changes due to nasal mucosal pathology [29]. In our study, 4PR was correlated with PNIF (r = −0.31, *p* = 0.0016) and AR (r = −0.5, *p* < 0.001), while PNIF and AR were not significantly correlated with each other (r = 0.18, *p* = 0.077) (Figure 1). Unilateral measurements showed a similar picture, apart from the significant correlation between lPNIF and lAR and between rPNIF and rAR (r = 0.26, *p* = 0.0087 and r = 0.32, *p* = 0.0011, respectively) (Figure 2 and Figure 3).

When looking at the correlation between the objective measurements and the nasal symptoms, only PNIF was significantly correlated with both SNOT-22 and VAS-NO. These findings corroborate what reported by Volstad and coworkers [30] who found a clear, although weak, correlation between unilateral PNIF and both VAS-NO and SNOT-22.

When we divided the population into subjects reporting an absent or mild nasal obstruction (VAS-NO < 5) and those referring a moderate-to-severe nasal obstruction (VAS-NO ≥ 5), PNIF values were significantly lower in the latter group. Equally, 4PR showed a significant increase of nasal resistances in the group of patients with higher VAS values (Figure 4a). Once we grouped the subjects into those reporting sinonasal symptoms (SNOT-22 ≥ 22) and those who did not (SNOT-22 < 22), none of the methods that were under evaluation showed a significant difference between the two groups (Figure 4b). As expected, VAS-NO showed better correlations with the objective measurement of nasal airflow than the SNOT-22 did. This can easily be explained by the fact that SNOT-22 has been designed to measure the quality of life in patients with chronic rhinosinusitis [1] and, thus, is less specific in evaluating nasal obstruction on its own. Moreover, VAS-NO offers an easy, quick, reproducible and quantifiable way to evaluate the severity of patients’ nasal obstruction symptom [12,22,30]. 

Whether 4PR, that surely provides supplementary information when it is compared to a normal RM, is more sensitive than classic RM in the diagnosis and characterization of nasal obstruction is still matter of debate. In fact, in a previous study, our research group found a significant correlation between RM and subjective nasal obstruction sensation (r = 0.22, *p* = 0.013) in both healthy and pathological noses [31]. In the present paper, no significant correlation between 4PR and VAS-NO (r = 0.2, *p* = 0.052) was observed; nonetheless, the *p*-value that was obtained was very close to the 5% level of significance. Recently, 4PR and RM have been shown to not significantly differ in terms of the outcomes that they produce [32], but these experiments were conducted in vitro, using artificial nose models [5]. Given the conflicting results that are found in the literature, further studies are needed to clarify whether 4PR is superior to classic RM in the assessment of nasal obstruction. Our findings also show that AR is the method that correlates to a lesser degree with both reported nasal obstruction (VAS-NO) and general sinonasal symptoms (SNOT-22). On the other hand, AR demonstrated a reasonable correlation with 4PR; therefore, 4PR should be preferred to AR in the clinical practice to evaluate nasal obstruction.

### Strength and Limitations

To our knowledge, this is the first study that has compared PNIF, AR and 4PR to each other and with the patients’ reported nasal obstruction sensation (measured by means of VAS-NO) and their sinonasal symptoms (measured by using SNOT-22). Furthermore, our subjects were enrolled consecutively, and all the measurements were performed by the same operator. The relatively low number of patients that were enrolled could be considered a limitation of the study. However, the study was powerful enough to confidently conclude that where a lack of correlation was observed, this was more likely to be a genuine result rather than a limitation of the test’s discriminatory power. The NOSE score, a PROM that was specifically validated for the subjective measurement of nasal obstruction, was not used on our volunteers, which can be considered another limitation. Finally, although a stronger correlation has been reported between unilateral nasal resistance (measured by RM) and ipsilateral VAS [33], we were not able to confirm this assumption because we did not collect data on unilateral VAS-NO (i.e., for a single nostril). 

## 5. Conclusions

PNIF, AR and RM assess different aspects of the nasal airways and in the present paper it was shown that they were correlated with each other. Although the correlations that we found were mainly weak or moderate, our results confirm that a correlation among the three different tools, despite not linear, still exists [34]. In our study, PNIF was better correlated with both nasal obstruction and the overall sinonasal symptoms when compared to AR and 4PR. Therefore, being inexpensive, fast, portable and reliable in determining nasal changes [35], it could be considered the preferred method to use for the routine objective assessment of nasal obstruction [36]. We would have found stronger correlations between the objective methods for the measurement of nasal obstruction (namely PNIF, AR and 4PR) and patients’ reported nasal obstruction (VAS-NO). Nevertheless, since nasal obstruction is a multifactorial symptom that is influenced also by the patients’ psychological status, we believe that the significances found in this study, although weak-moderate, are still important as they demonstrate an overall correlation between the nasal obstruction symptom and some of the objective methods for the measurement of nasal obstruction, in particular PNIF. It should be kept in mind that the measurement of nasal obstruction is particularly important in those patients with reported symptoms that appear to be conflicting. In these patients, in fact, the objective measurement of nasal obstruction can lead to insights into the nature of the symptoms, thus helping to differentiate between normal and pathological noses.

Future studies should include computational fluid dynamics in this comparison analysis, considering that a degree of correlation with RM has also been demonstrated [37].

## Figures and Tables

**Figure 1 jpm-12-01513-f001:**
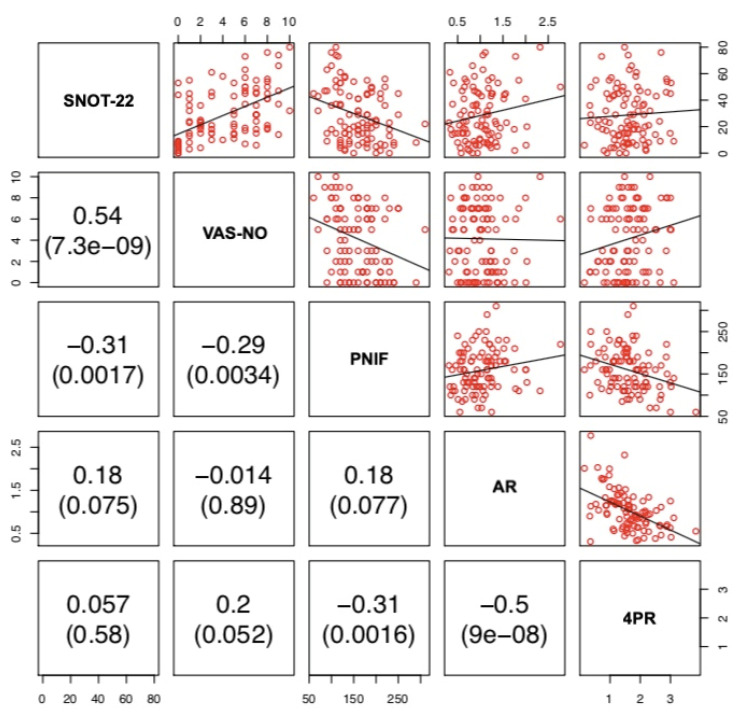
Correlation between SNOT-22, VAS-NO, bilateral PNIF, AR and 4PR. PNIF: Peak Nasal Inspiratory Flow; AR: acoustic rhinometry; 4PR: 4-Phase Rhinomanometry; SNOT-22: Sinonasal Outcome Test-22; VAS-NO: Visual Analogue Scale for Nasal Obstruction.

**Figure 2 jpm-12-01513-f002:**
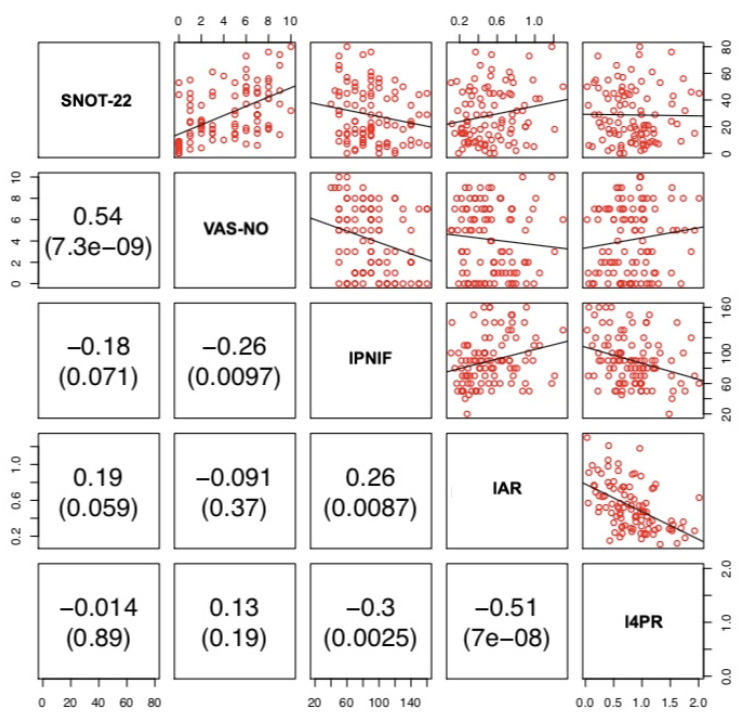
Correlation between SNOT-22, VAS-NO, lPNIF, lAR and l4PR. lPNIF: left Peak Nasal Inspiratory Flow; lAR: left acoustic rhinometry; l4PR: left 4-Phase Rhinomanometry; SNOT-22: Sinonasal Outcome Test-22; VAS-NO: Visual Analogue Scale for Nasal Obstruction.

**Figure 3 jpm-12-01513-f003:**
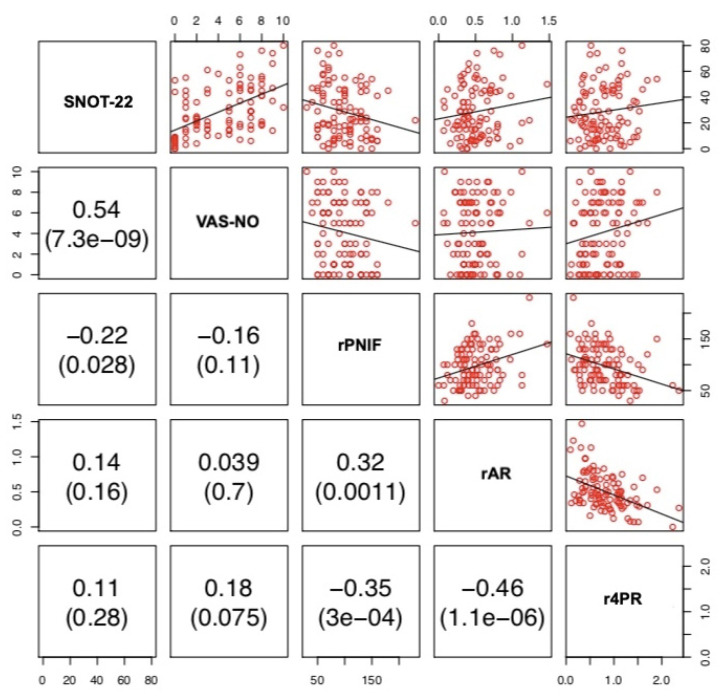
Correlation between SNOT-22, VAS-NO, rPNIF, rAR and r4PR. rPNIF: right Peak Nasal Inspiratory Flow; rAR: right acoustic rhinometry; r4PR: right 4-Phase Rhinomanometry; SNOT-22: Sinonasal Outcome Test-22; VAS-NO: Visual Analogue Scale for Nasal Obstruction.

**Figure 4 jpm-12-01513-f004:**
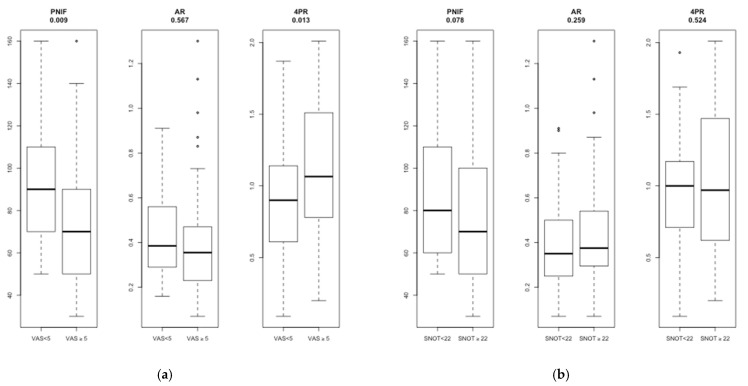
Box-plot of PNIF, AR and 4PR compared with VAS-NO (**a**) and SNOT-22 (**b**). PNIF: Peak Nasal Inspiratory Flow; AR: acoustic rhinometry; 4PR: 4-Phase Rhinomanometry.

**Table 1 jpm-12-01513-t001:** Patients’ main clinical characteristics and values of the objective measurements of the nasal function and evaluation of symptoms.

	Subjects (*n* = 101)	SNOT-22 ≥ 22 (*n* = 53)	VAS ≥ 5 (*n* = 50)	SNOT-22 ≥ 22 and VAS ≥ 5 (*n* = 36)
Age, median [P25–P75], yr	45 [27–58]	47 [30–59]	42 [27–56]	46 [29–59]
Sex, No (%)				
Female	43 (42.6%)	28 (52.8%)	19 (38%)	17 (47.2%)
Male	58 (57.4%)	25 (47.2%)	31 (62%)	19 (52.8%)
Height, median [P25–P75], m	1.72 [1.64–180]	1.70 [1.60–1.80]	1.73 [1.63–1.80]	1.70 [1.60–1.80]
BMI, median [P25–P75], Kg/m^2^	23.8 [21.54–26.35]	23.8 [21.6–26.6]	24.1 [21.6–27]	24.1 [21.6–26.3]
Allergy, No (%)	50 (49.5%)	28 (52.8%)	28 (56%)	19 (52.8%)
Asthma, No (%)	24 (23.8%)	13 (24.5%)	11 (22%)	7 (19.4%)
Smoke, No (%)	13 (12.9%)	3 (5.7%)	5 (1%)	2 (5.5%)
Reason for attendance, No (%)				
CRSsNP	16 (15.8%)	11 (20.7%)	11 (22.0%)	6 (16.0%)
CRSwNP	30 (29.7%)	15 (28.3%)	8 (1.6%)	9 (25.0%)
Nasal septal deviation	26 (25.7%)	15 (28.3%)	17 (34.0%)	12 (33.3%)
CRS + nasal septal deviation	8 (7.9%)	5 (9.4%)	7 (1.4%)	4 (11.1%)
Other	21 (20.8%)	6 (11.3%)	6 (1.2%)	4 (11.1%)
Nasal medications, No (%)				
Only steroid spray	16 (15.8%)	9 (17.0%)	10 (20.0%)	6 (16.7%)
Only douches (normal saline)	13 (12.9%)	7 (13.2%)	7 (14.0%)	6 (16.7%)
Steroid and douches (normal saline)	26 (25.7%)	13 (24.5%)	10 (20.0%)	7 (19.4%)
Previous nasal surgery, No (%)				
ESS	25 (24.8%)	13 (24.5%)	9 (18.0%)	6 (16.7%)
Septoplasty	10 (9.9%)	6 (11.3%)	5 (10.0%)	5 (13.9%)
Turbinoplasty	2 (1.9%)	0 (0.0%)	0 (0.0%)	0 (0.0%)
PNIF, median [P25–P75], L/min	160 [120–190]	150 [110–190]	150 [110–180]	125 [110–180]
4PR, median [P25–P75], Pa/cm^3^·s	1.59 [1.14–1.64]	1.59 [1.04–1.99]	1.74 [1.32–2.12]	1.74 [1.27–2.12]
MCA, median [P25–P75], cm^2^	1 [0.67–1.24]	1.03 [0.77–1.26]	1.02 [0.65–1.17]	0.94 [0.69–1.20]
SNOT-22, median [P25–P75]	24 [12–45]	45 [32–53]	37 [20–50]	45 [35–54]
VAS-NO, median [P25–P75]	5 [1–7]	6 [3–7]	7 [6–8]	7 [6–8]

BMI: Body Mass Index; PNIF: Peak Nasal Inspiratory Flow; MCA: Minimal Cross-sectional Area; 4PR: 4-Phase Rhinomanometry; SNOT-22: Sinonasal Outcome Test-22; VAS-NO: Visual Analogue Scale for Nasal Obstruction.

## Data Availability

The data that support the findings of this study are available from the corresponding author upon reasonable request.

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
