# Peer review of "Correlations between Peak Nasal Inspiratory Flow, Acoustic Rhinometry, 4-Phase Rhinomanometry and Reported Nasal Symptoms"

_jpm, 2022, doi:10.3390/jpm12091513_

Round 1
Reviewer 1 Report
This manuscript is a correlation study between objective tools to evaluate nasal patency, including 4-Phase Rhinomanometry (4PR), acoustic rhinometry (AR) and peak nasal inspiratory flow (PNIF), and subjective scales, such as Visual Analogue Scale for nasal obstruction (VAS-NO) and Sino-Nasal Outcome Test (SNOT-22).
It is an interesting and well-written paper.
I recommend its publication.
However, it needs major improvements:
1.- Mat & Method
Sample size calculations are missing.
In lines 98-101, the authors state: “Bravais-Pearson correlation coefficient was used to measure the relation between the different indicators. In order to adjust for sex, age, BMI, allergy, asthma and smoking, different linear models were fitted using PNIF, AR, 4PR, VAS-NO, SNOT-22 as response variable; correlations among residuals were used as measures of adjusted correlation.”
Please, explain when such correlations are relevant to this clinical study, rather than significant.
See van Egmond et al. (2017) page 128: “Coefficients less than -0.70 or greater than 0.70 were defined as strong correlation; coefficients in the range of -0.70 to -0.30 or in the range of 0.30 to 0.70 expressed moderate correlation; and coefficients greater than -0.30 but less than 0.30 were considered as indicators of weak correlation”.
2.- Results
In lines 118-124, only “VAS-NO and SNOT-22 showed a positive correlation (r=0.54, p<0.001)”. All other were below 0.5.
Please, explain it detail. For instance: line 137-138: “we found a significant negative correlation with PNIF (r=-0.29, p=0.0034) and lPNIF (r=-0.26, p=0.0097)”.
Most results of this manuscript only trends, and should be mentioned so during discussion and conclusions.
3.- Conclusions
In lines 243-245, the authors state: “In the present study, PNIF better correlated with both nasal obstruction and overall sinonasal symptoms when compared to AR and 4PR. Therefore, being inexpensive, fast, portable and reliable in determining nasal changes”.
This sentence should be removed, because it is not a correct conclusion from this study, as it is only based on a trend.
Author Response
According to reviewer 1 suggestions:
1. Sample size calculations are missing.
According to reviewer suggestion, a sample size calculation has been done and included in the methods and discussion sections (see lines 106-107, 108 and 227-229)
In lines 98-101, the authors state: “Bravais-Pearson correlation coefficient was used to measure the relation between the different indicators. In order to adjust for sex, age, BMI, allergy, asthma and smoking, different linear models were fitted using PNIF, AR, 4PR, VAS-NO, SNOT-22 as response variable; correlations among residuals were used as measures of adjusted correlation.”
Please, explain when such correlations are relevant to this clinical study, rather than significant.
See van Egmond et al. (2017) page 128: “Coefficients less than -0.70 or greater than 0.70 were defined as strong correlation; coefficients in the range of -0.70 to -0.30 or in the range of 0.30 to 0.70 expressed moderate correlation; and coefficients greater than -0.30 but less than 0.30 were considered as indicators of weak correlation”>.
We thank the reviewer for the suggestion which allowed us to highlight important aspects of the study. As reported in the introduction, several studies have compared these three methods, but results are conflicting. Some authors (Yepes-Nunez et al, 2013; Pendolino et al, 2018) found that usually the correlation between the different methods for the measurement of nasal obstruction is moderate (r>0.3), while others did not find any significance. The same is true if we consider the correlations between objective and subjective methods for the measurement of nasal obstruction (Fokkens and Hellings, 2014; Simola et al, 1997). In this paper (see the conclusions) we underlined that the correlations found were moderate and, in many cases, weak. Nevertheless, as mentioned in the discussion, nasal obstruction is a multifactorial symptom consequence of the intranasal anatomical status, but also influenced by the autonomic nervous system and other different physiological and pathological factors. Furthermore, its estimation can be influenced by the patients' psychological status or expectations. For all these reasons the correlations between the measurement of nasal obstruction and the reported nasal obstruction symptom, when significant, is often moderate to weak. Nevertheless, we believe that these weak-to-moderate significances are still important because they confirm a general correlation between the nasal obstruction symptom and some of the objective methods for the measurement of nasal obstruction. The measurement of nasal obstruction is particularly important in those patients who have airway testing results that do not match symptoms. In these patients in particular, the objective measurement of nasal obstruction can lead to insight into the nature of the symptoms. We added some of these comments in the conclusions (lines 243-254).
2. In lines 118-124, only “VAS-NO and SNOT-22 showed a positive correlation (r=0.54, p<0.001)”. All other were below 0.5. Please, explain it detail. For instance: line 137-138: “we found a significant negative correlation with PNIF (r=-0.29, p=0.0034) and lPNIF (r=-0.26, p=0.0097)”.
Most results of this manuscript only trends, and should be mentioned so during discussion and conclusions>.
We thank the reviewer for the suggestion. We have added the requested information in the results (lines 126-134). Furthermore, we deleted all the statistical trends in the entire text and considered only statistically significant results (p<0.05). Abstract, result, discussion and conclusion have been changed accordingly.
3. In lines 243-245, the authors state: “In the present study, PNIF better correlated with both nasal obstruction and overall sinonasal symptoms when compared to AR and 4PR. Therefore, being inexpensive, fast, portable and reliable in determining nasal changes”.
This sentence should be removed, because it is not a correct conclusion from this study, as it is only based on a trend>.
We are sorry, but the correlation between PNIF and SNOT-22 was significant (p=0.0017). The same happened when looking at the correlation between PNIF and VAS-NO (p=0.0034), while the correlations between AR and 4PR with SNOT-22 and VAS-NO are not significant or are statistical trends. The sentence seems correct and has been kept in the manuscript.
Reviewer 2 Report
This is a study of correlation among three important test of nasal obstruction (PNIF, AR and 4PR), and also correlation with the VAS-NO. The methodology is fine. There are some comments about the conclusion and statistical interpretation.
1) The Pearson's correlation (r) is usually interpreted as mild (<0.3), moderate (0.3 to <0.7) and strong (>0.7). So the authors should considered justify the interpretation of the key finding of this study, especially the line#25-27.
2) The terminology of 'marginal correlation' (line#27) and 'roughly correlated' (line#32) should be justified whether they are universally used or not.
3) Correlation of PNIF vs VAS-NO (r = -0.29) vs VAS-NO with 4PR (r = 0.2) is not much different. When the last sentence of conclusion as '... PNIF showed the best correlations...' , it seems rather over statement.
4) The sentence in line#103 , '....p value 0.05 <p<0.15 was considered a trend towards a significant correlation.' should be cited whether it is a standard statistical concept or the author's opinion.
Author Response
Padova, 10/09/2022
Dear Editor,
we are submitting the revised manuscript “Correlations between peak nasal inspiratory flow, acoustic rhinometry, 4-phase rhinomanometry and reported nasal symptoms” which was edited according to reviewers’ comments and which we would like you to consider for publication in “J. Pers. Med”.
We would like to thank the reviewer panel for reconsidering of the article.
The changes in the manuscript have been highlighted and a list of all changes with a point-by-point reply to reviewers’ comments is enclosed.
Best regards,
Giancarlo Ottaviano, MD, PhD
Correspondence to:
Giancarlo Ottaviano, MD
Dept Neurosciences, Otolaryngology Section, University of Padova
Via Giustiniani 2, 35100 Padova, Italy; giancarlo.ottaviano@unipd.it;
fax +39 (0)49 8213113
According to reviewer 2 suggestions:
- The Pearson's correlation (r) is usually interpreted as mild (<0.3), moderate (0.3 to <0.7) and strong (>0.7). So the authors should considered justify the interpretation of the key finding of this study, especially the line#25-27.
We thank the reviewer for the suggestion. The lines indicated have been changed accordingly (see also the answer to reviewer 1 comment 1.b) (lines 243-254).
2. The terminology of 'marginal correlation' (line#27) and 'roughly correlated' (line#32) should be justified whether they are universally used or not. We thank the reviewer for the suggestion.
We deleted both “marginal” and “roughly” correlated. Furthermore, we considered significant the results with p-value <0.05. The text has been changed accordingly (see abstract).
3. Correlation of PNIF vs VAS-NO (r = -0.29) vs VAS-NO with 4PR (r = 0.2) is not much different. When the last sentence of conclusion as '... PNIF showed the best correlations...' , it seems rather over statement.
We thank the reviewer for the suggestion, after not considering the results showing a statistical trend, we found that only PNIF correlated with VAS-NO. In the abstract, the sentence “PNIF showed the best correlations….” Was changed with “PNIF showed a significant correlation (weak- moderate) with reported nasal symptom scores (line 33).
4. The sentence in line#103 , '....p value 0.05 <p<0.15 was considered a trend towards a significant correlation.' should be cited whether it is a standard statistical concept or the author's opinion.
According to the suggestion, we decided to not consider the results with a p-value 0.05<p<0.15 as a statistical trend or marginally significant. Abstract, result, discussion and conclusion have been corrected accordingly.
Round 2
Reviewer 1 Report
All requests have been answered. Accepted!